# Pterostilbene Reverses Epigenetic Silencing of Nrf2 and Enhances Antioxidant Response in Endothelial Cells in Hyperglycemic Microenvironment

**DOI:** 10.3390/nu16132045

**Published:** 2024-06-27

**Authors:** Kannan Harithpriya, Kumar Ganesan, Kunka Mohanram Ramkumar

**Affiliations:** 1Department of Biotechnology, School of Bioengineering, SRM Institute of Science and Technology, Kattankulathur 603203, India; harithpriya2201@gmail.com; 2School of Chinese Medicine, Li Ka Shing Faculty of Medicine, The University of Hong Kong, 10 Sassoon Road, Pokfulam, Hong Kong 999077, China

**Keywords:** pterostilbene, polyphenol, Nrf2, HDACs, hyperglycemia, endothelial cells

## Abstract

The epigenetic regulation of nuclear factor erythroid 2-related factor 2 (Nrf2), a pivotal redox transcription factor, plays a crucial role in maintaining cellular homeostasis. Recent research has underscored the significance of epigenetic modifications of Nrf2 in the pathogenesis of diabetic foot ulcers (DFUs). This study investigates the epigenetic reversal of Nrf2 by pterostilbene (PTS) in human endothelial cells in a hyperglycemic microenvironment (HGM). The activation potential of PTS on Nrf2 was evaluated through ARE-Luciferase reporter assays and nuclear translocation studies. Following 72 h of exposure to an HGM, mRNA expression and protein levels of Nrf2 and its downstream targets NAD(P)H quinone oxidoreductase 1 (NQO1), heme-oxygenase 1(HO-1), superoxide dismutase (SOD), and catalase (CAT) exhibited a decrease, which was mitigated in PTS-pretreated endothelial cells. Epigenetic markers, including histone deacetylases (HDACs class I–IV) and DNA methyltransferases (DNMTs 1/3A and 3B), were found to be downregulated under diabetic conditions. Specifically, Nrf2-associated HDACs, including HDAC1, HDAC2, HDAC3, and HDAC4, were upregulated in HGM-induced endothelial cells. This upregulation was reversed in PTS-pretreated cells, except for HDAC2, which exhibited elevated expression in endothelial cells treated with PTS in a hyperglycemic microenvironment. Additionally, PTS was observed to reverse the activity of the methyltransferase enzyme DNMT. Furthermore, CpG islands in the Nrf2 promoter were hypermethylated in cells exposed to an HGM, a phenomenon potentially counteracted by PTS pretreatment, as shown by methyl-sensitive restriction enzyme PCR (MSRE-qPCR) analysis. Collectively, our findings highlight the ability of PTS to epigenetically regulate Nrf2 expression under hyperglycemic conditions, suggesting its therapeutic potential in managing diabetic complications.

## 1. Introduction

Impaired wound closure in diabetic complications, particularly diabetic foot ulcer (DFU), involves a complex interplay of neuropathic, biochemical, vascular, and immune mechanisms [1,2]. Prolonged hyperglycemia is known to disrupt endothelial cell function, leading to vasodilation, impaired angiogenesis, decreased cell proliferation, and ultimately contributing to delayed wound healing in DFU [3,4]. Endothelial cells play a crucial role in regulating inflammation during the wound-healing process. Factors such as hyperglycemia, insulin resistance, increased oxidative stress, inflammation, and hypertension, contribute to endothelial cell damage [5]. Evidence suggests that excessive reactive oxygen species (ROS) production contributes to impaired wound healing in diabetic wounds [6,7]. The accumulation of ROS disrupts the ability of vascular endothelial cells to regulate the passage of circulatory cells and macromolecules to the tissue, leading to delayed healing [8]. Suppressing ROS accumulation is crucial for maintaining cellular homeostasis and is achieved through triggering antioxidant-driven response elements [9].

Nuclear factor erythroid 2-related factor 2 (Nrf2), a master regulator in cellular defense signaling, is compromised in diabetic conditions. Nrf2 induces phase II detoxifying and antioxidant response elements (AREs), thereby acting as a key regulator of redox homeostasis [10]. It has been well established that the Nrf2-mediated antioxidant defense pathway provides vascular protection against diabetes and its complications [11,12]. Studies by Min Long et al. demonstrated the essential role of Nrf2 in diabetic wound healing using Nrf2^−/−^ diabetic mice, while its involvement in non-diabetic wound healing has also been explored previously [13].

Although the regulation of Nrf2 by lncRNA (long non-coding RNA) and miRNA (microRNA) is well established, it is important to note the significant role of histone modification and methylation in this process. Recent discoveries underscore the significance of “epigenetic erasers” like histone deacetylases (HDACs) and “epigenetic writers” like DNA methyltransferases (DNMTs) in disease progression due to their role in regulating gene expression. HDACs play critical roles in essential cellular functions, including homeostasis [14,15]. DNMT-mediated CpG methylation impairs transcription factor activity, leading to gene silencing [16]. Both epigenetic markers tend to increase cellular ROS production, causing cellular stress. HDAC enzymes modulate oxidative stress by promoting the expression of NADPH oxidase (Nox) [17]. Similarly, DNMTs induce changes in mitochondrial function, which can alter ROS production, leading to delayed wound healing [18]. The epigenetic modifications of Nrf2, especially DNA hypermethylation, have been documented in various diseases, such as cancers and pre-eclampsia [19]. Additionally, studies have shown that DNA demethylation promotes the Nrf2 cell signaling pathway, potentially enhancing the antioxidant system to counteract the development of Alzheimer’s disease [20].

Existing evidence highlights the growing interest in the small-molecule-mediated activation of Nrf2. Specifically, pharmacological compounds that activate Nrf2 have demonstrated cytoprotective effects under various stress conditions [21]. We have previously documented the activation of Nrf2 and its associated biological functions by morin, mangiferin, rosalic acid, and vitexin [22,23,24,25]. Pterostilbene (PTS), a compound belonging to the stilbene family and structurally similar to resveratrol, has garnered attention for its potent pharmacological activity against stress, as well as other numerous therapeutic advances in diseases including cardiovascular, metabolic, neurological, and hematological diseases. Although berries contribute to the main source of PTS, other sources like grape leaves, peanuts, red sandalwood, and Indian kino contain PTS in abundance [21]. PTS has been shown to inhibit oxidative stress, leading to the increased production of antioxidant defense enzymes [14]. Moreover, PTS has the potential to reduce the production of the superoxide anion and hydrogen peroxide, thereby inhibiting disease progression [26]. PTS has been shown to inhibit various disease pathogeneses due to its anti-inflammatory, anti-carcinogenic, and antioxidant properties, thus improving normal cell function and contributing to malignant cell inhibition [27,28].

Our previous research demonstrated that PTS treatment restored dysregulated Nrf2 pathway in diabetic conditions in vitro [29]. Additionally, PTS treatment attenuated inflammation in macrophages due to cellular stress in non-healing wound conditions [30]. Another study from our laboratory showed that PTS, as a potent Nrf2 activator, protects pancreatic β-cells against oxidative stress [31]. With its anti-inflammatory, antioxidant, and antidiabetic properties, PTS has shown promising effects against various disease pathologies, including DFU. In our current study, we aim to investigate the effect of PTS on the epigenetic reactivation of Nrf2 in endothelial cells in a hyperglycemic microenvironment. The findings have the potential to be translated into clinical applications, such as developing PTS-based treatments or combination therapies for diabetic patients.

## 2. Materials and Methods

### 2.1. Materials

Pterostilbene (PTS) was purchased from Cayman chemical company, Ann Arbor, MI, USA (Cat No: 13000, Purity ≥ 98%). The mouse recombinant cytokines such as TNF-α (tumor necrosis factor-α), IFN-γ (interferon-γ), and IL-1β (interleukin-1β) were purchased from Abcam, UK. All the cytokines were dissolved in sterile saline-buffered phosphate (PBS) with 0.1% BSA and stored at −20 °C. The concentration of DMSO for dissolving PTS was restricted to a final concentration of 0.1% (*v*/*v*).

### 2.2. Cell Culture Conditions

Human endothelial cells (EA.hy929) were cultured in Dulbecco Modified Eagle Medium (DMEM) (Hi-media, Thane, India) at 37 °C in a 5% CO_2_-humidified incubator supplemented with 10% FBS (GIBCO, Life Technologies, Carlsbad, CA, USA). Once the cells reached confluency, they were trypsinized and passaged. To establish the hyperglycemic microenvironment (HGM), the cells were initially serum-starved in DMEM with 1% FBS for 3 h. Subsequently, they were exposed to a final concentration of 33.3 mM glucose along with 20 ng/mL of IL-1β, TNF-α, and IFN-γ for a duration of 24 h. The high glucose and cytokine cocktail were used in combination to mimic the human diabetic wound biological hyperglycemic microenvironment.

### 2.3. Cell Viability Assay

The endothelial cells were seeded onto a 96-well plate at a density of 20,000 cells per well. To determine the non-toxic concentrations of PTS and the effect of the HGM, the cells were treated as follows: They were exposed to PTS alone (0–100 μM) for 24 h. Following the experiments, the cells were incubated with a 0.1% Alamar blue solution (Hi-media, Thane, India) for 4 h, and absorbance was measured at 570 nm and 600 nm using a microplate reader (TECAN, Männedorf, Switzerland).

To explore the protective effect of PTS against HGM-induced cell death, endothelial cells were pretreated with PTS at concentrations ranging from 0 to 25 μM for 24 h. Subsequently, the cells were exposed to an HGM for an additional 24 h before conducting the Alamar blue assay as described previously.

### 2.4. Luciferase Reporter Assay

The activation potential of PTS was evaluated using the ARE-luciferase reporter assay system. The GST1 and hNQO1-ARE-Luc construct, generously provided by Donna D. Zhang from the College of Pharmacy, University of Arizona, Tucson, AZ, USA, was used in this study. Briefly, endothelial cells were transfected with the ARE-luciferase vector (500 ng) using Lipofectamine 3000 reagent (Invitrogen, Waltham, MA, USA). Following a 6 h incubation period, the medium was replaced, and the cells were exposed to various concentrations of PTS (0–10 µM) for 8 h. Subsequently, the cells were lysed with cell lysis buffer (Promega, Madison, WI, USA), and the luciferase activity was measured using a luminometer (Promega, Madison, WI, USA). The luciferase activity was quantified as fold change relative to the scrambled control, considered as the baseline for reporter gene activation [32].

### 2.5. Quantitative Real-Time PCR Analysis

After the treatment conditions, the cells were washed with 1× PBS, harvested using trypsin–EDTA solution (Lonza, Basel, Switzerland), and subjected to RNA isolation using TRizol Reagent (RNAiso Plus, Takara Bio, Kusatsu, Japan). The concentration of isolated RNA was assessed using a Nano quant (TECAN, Männedorf, Switzerland), with RNA samples exhibiting a purity of 2.0 serving as templates for cDNA synthesis via the PrimeScript RT Reagent kit (Takara, Kusatsu, Japan), following the manufacturer’s instructions. The expression of Nrf2, its downstream targets (NQO1, CAT, SOD2, and HO-1), HDACs (HDAC1, 2, 3, and 4), and DNMTs (DNMT1, 3A, and 3B) were quantified using 2*^−^*^ΔΔCt^, normalized to the GAPDH housekeeping gene (Table 1) [33].

### 2.6. Nrf2 Activation Potential of PTS by Immunoblotting

To study the effect of PTS on Nrf2 translocation, nuclear and cytoplasmic extracts were prepared using the Nuclear Extraction Kit (Abcam, Cambridge, UK) following the manufacturer’s protocol. In brief, cells were homogenized in pre-extraction buffer using a homogenizer and subsequently incubated on ice for 15 min. Following this, the homogenates were centrifuged at 10,000× *g* for 10 min at 4 °C. The resulting supernatant was collected as the cytoplasmic fraction. Pellets containing nuclei were then suspended in extraction buffer supplemented with protease inhibitors as per the manufacturer’s instructions. After thorough vortexing for 40 min with a 1 min break every 10 min, the samples were centrifuged at 16,000× *g* for 15 min at 4 °C, and the supernatant was collected as the nuclear fraction. Parallelly, the cells were lysed using Radio Immune Assay Buffer (RIPA, Sigma-Aldrich, Saint Louis, MO, USA) and subjected to centrifugation at 16,000× *g* for 15 min to recover the protein fraction. Protein concentrations were determined using the Bradford assay (Bio-Rad, Hercules, CA, USA), and the samples were subsequently subjected to Western blot analysis.

For Western blot analysis, equal amounts of protein (40 μg) were separated by SDS-PAGE and then transferred onto nitrocellulose (NC) membranes using a semi-dry transfer unit (Bio-Rad, Hercules, CA, USA). Subsequently, the NC membranes were blocked with 3% BSA and incubated with primary antibodies against Nrf2 (1:500, A0674, Abclonal, Woburn, MA, USA), β-actin (1:1000 dilution; sc-47778, Santa Cruz, CA, USA), and Lamin B1 (1:10,000, ab16048, Abcam, Cambridge, MA, USA). After an overnight incubation with the primary antibodies, the membranes were washed and then incubated with a suitable HRP-conjugated secondary antibody, the blots were visualized using the enhanced chemiluminescence kit (ECL kit, Bio-Rad, Hercules, CA, USA) and Chemidoc system (Vilber Lourmat, Collégien, France), and the analysis of the image was carried out using ImageJ software (v. 1.53t, NIH, Bethesda, MD, USA). Concurrently, along with Nrf2, the downstream target SOD2 (1:500 dilution, Sc-30080, Santa Cruz, CA, USA) was also examined in total cell homogenate [33].

### 2.7. Bisulfite Conversion and Primer Designing

To validate the effect of PTS on epigenetic reactivation through the methylation of Nrf2, the bisulfite conversion process was carried out. Following treatment, cells were harvested, and DNA isolation was performed using a commercially available kit (Qiagen, Hilden, Germany) according to the manufacturer’s instructions. The purity and quantity of the isolated DNA were determined using a Nano quant (TECAN, Männedorf, Switzerland), with DNA samples exhibiting a purity of 1.8 considered suitable for this protocol. Bisulfite conversion was carried out using the commercially available Epitect Bisulfite kit (Qiagen, Hilden, Germany) following the manufacturer’s protocol [34].

The first step in primer design involved identifying and retrieving the Nrf2 promoter region from the UCSC browser. The FASTA sequence was then analyzed to identify CpG islands (CG-rich regions), which are prone to methylation, using the EMBOSS CpG plot tool. Considering the bisulfite-converted sequence as a template, primers for the Nrf2 target were designed using an online primer-designing tool.

### 2.8. Status of Nrf2 Promotor CpG Island Methylation and Its Reversal by PTS

To assess the potential of PTS in reversing the methylation percentage in the Nrf2 CpG promoter region, we utilized the One-step qMethyl kit (Zymo-Research, Tustin, CA, USA). According to the kit protocol, the bisulfite-converted DNA was diluted to a concentration of 4 ng/µL and divided into two groups: the reference and the test reaction. In the test reaction, the bisulfite-converted DNA was digested for two hours with methylation-sensitive restriction enzymes (MSREs) specific to the Nrf2 promoter CpG island. Both the test and reference reactions were then amplified using qRT-PCR with SYTO^®^9 fluorescent dye. The percentage of methylation was analyzed using the following formula [35]:ΔCt = average Ct value from test reaction − the average Ct value from reference reaction

### 2.9. Statistical Analysis

All the experiments were carried out in triplicates and the results are represented as mean ± SEM (standard error mean), and *p* ≤ 0.05 was considered statistically significant. The *p*-values were calculated using the student’s *t*-test followed by one-way ANOVA using Tukey’s multiple comparison test with GraphPad Prism (version 8.01). The Western blot images were quantified using ImageJ software and analyzed using GraphPad Prism. Pearsons’s correlation was performed on the treatment groups using SPSS software (V.20.0) [30].

## 3. Results

### 3.1. Cytotoxicity of PTS in EA.hy929 Cells

To evaluate the cytotoxicity of PTS, endothelial cells were exposed to various concentrations of PTS (0–100 µM) for 24 and 48 h, and the viability was determined using the Alamar blue assay. As depicted in Figure 1a, the cytotoxic effects were observed starting from a concentration of 25 µM PTS after both 24 and 48 h of treatment. Therefore, the concentration of PTS was limited to 25 µM for subsequent experiments. Considering the treatment duration, no significant cytotoxicity was observed in cells treated with up to 25 µM PTS for 24 h. However, after 48 h of exposure, cytotoxicity was evident at 25 µM PTS concentration, with minimal toxicity observed at 10 µM PTS concentration. Based on this analysis, the exposure to PTS of endothelial cells was restricted to 24 h for further experiments.

To assess the cytoprotective effect of PTS in a hyperglycemic microenvironment, endothelial cells were pretreated with PTS (0–25 µM) for 24 h followed by exposure to the hyperglycemic microenvironment for an additional 24 h. Subsequently, the Alamar blue assay was performed. As depicted in Figure 1b, we observed a 50% reduction in the viability of HGM-induced endothelial cells. However, cells pretreated with PTS showed increased viability at 5 and 10 μM concentrations compared to the untreated cells. This indicates a profound protective property of PTS against HGM-induced cytotoxicity in endothelial cells.

### 3.2. Nrf2 Activation Potential by PTS on Endothelial Cells by Luciferase Reporter Assay

To investigate the dose-dependent activation of Nrf2 by PTS, we utilized a reporter-cell-based luciferase assay with the GST and hNQO1-ARE-Luc gene construct. As depicted in Figure 2, there was a significant increase in the expression of NQO1 and GST in endothelial cells treated with different concentrations of PTS. The maximum increase was observed at the 10 µM concentration after 24 h of exposure. Based on these results, the concentration of PTS was fixed at 10 µM, and the exposure duration was set to 24 h for further experiments. This concentration and exposure time were chosen to ensure optimal activation of Nrf2 with no cytotoxic effects.

### 3.3. Effect of Pterostilbene on Nrf2 Activation and Nuclear Translocation in Endothelial Cells in a Hyperglycemic Microenvironment

The activation of Nrf2 by PTS was confirmed through a nuclear translocation experiment using Western blot analysis. The expression of Nrf2 in cytosolic fractions was normalized with β-actin, while the expression in nuclear fractions was normalized with Lamin B1 protein. As illustrated in Figure 3, the nuclear/cytosolic protein expression ratio of Nrf2 was elevated in endothelial cells pretreated with PTS (0–10 µM) in an HGM, whereas it was lower in untreated cells. A dose-dependent increase in Nrf2 protein expression was observed when compared to untreated cells. These results support the findings of the reporter-cell-based assay, indicating the translocation of Nrf2 in cells pretreated with PTS (10 µM).

### 3.4. Pterostilbene Modulates Nrf2 Expression in Endothelial Cells in an HGM

The effect of PTS on modulating Nrf2 expression was assessed through qRT-PCR and Western blot analysis. Figure 4a depicts the gene expression analysis, revealing significantly lower Nrf2 expression compared to untreated control cells. However, the pretreatment of HGM-induced endothelial cells with PTS (10 µM) led to increased gene expression, confirming the activation potential of PTS.

Similarly, protein expression was diminished in endothelial cells under HGM conditions compared to the control, as shown in Figure 4b. Notably, cells pretreated with PTS (10 µM) under HGM conditions exhibited increased Nrf2 expression, with a fold change of 1.8 (*p* < 0.001).

### 3.5. Pterostilbene Modulates the Expression of Nrf2 and Its Downstream Targets in Endothelial Cells in an HGM

The expression of Nrf2 downstream targets upon PTS exposure was assessed using qRT-PCR. In HGM-induced endothelial cells, a decrease in the expression of downstream targets including NQO1 (3-fold, *p* < 0.01), SOD2 (5-fold, *p* < 0.05), CAT (1.35-fold), and HO-1 (1.53-fold) was observed compared to untreated control cells. However, their expression was increased in cells pretreated with PTS (10 µM) compared to the control. Moreover, pretreatment with PTS (10 µM) resulted in the increased expression of Nrf2 downstream targets compared to the HGM group. Specifically, the expression of NQO1 and HO-1 showed a significant increase with an approximate fold change of 1.8, while the expression of SOD2 and CAT was elevated with an average fold change of 1.4 compared to the control (Figure 5a–d).

Analysis of protein expression (Figure 5e) of SOD2 in HGM-induced endothelial cells revealed lower expression under HGM conditions (1.65-fold, *p* < 0.01) compared to the control. This expression was reversed in cells pretreated with PTS under HGM conditions, with a fold change of 1.23 compared to cells exposed to an HGM alone.

### 3.6. Pterostilbene Modulates the Expression of HDACs in Endothelial Cells in an HGM

To understand the potential of PTS in modulating the expression of HDACs, a qRT-PCR experiment was performed across all treatment groups. HDAC 1, 2, 3, and 4, closely associated with Nrf2 based on earlier data, were selected for investigation regarding the effect of PTS on these epigenetic regulators under HGM conditions. The gene expression levels of HDACs revealed significant alterations in HGM-induced endothelial cells compared to the control. Specifically, HDAC1 (2.5-fold, *p* < 0.05), HDAC3 (1.8-fold, *p* < 0.05), and HDAC4 (4.2-fold, *p* < 0.001) were found to be upregulated in HGM-induced endothelial cells.

However, exposure to PTS (10 µM) inhibited the expression of HDAC 1, 3, and 4 compared to HGM-induced endothelial cells, as illustrated in Figure 6a,c,d. Conversely, the gene expression levels of HDAC2 were found to be lower in HGM-induced endothelial cells (1.72-fold, *p* < 0.01) compared to untreated cells. Treatment with PTS alleviated the expression level of HDAC2, with a fold change of 1.8 compared to untreated endothelial cells (Figure 6b).

### 3.7. Effect of Pterostilbene in Regulating the Expression of Epigenetic Writers DNA Methyltransferase (DNMTs)

To determine the effect of PTS in alleviating the expression of DNMTs, a qRT-PCR experiment was conducted across all treatment groups. The analysis revealed significant alterations in the expression of epigenetic writer DNMTs in HGM-induced endothelial cells compared to the control. As depicted in Figure 7, the expression levels of DNMTs DNMT1 (2.27-fold, *p* < 0.001), DNMT3A (2.46-fold, *p* < 0.01), and DNMT3B (2.8-fold, *p* < 0.05) were elevated in HGM-induced endothelial cells compared to the control. However, this increase in DNMT expression was attenuated in cells pretreated with PTS (10 µM) compared to HGM-induced endothelial cells.

### 3.8. Pterostilbene Reverses Nrf2 Promoter CpG Island Methylation

To assess the status of Nrf2 promoter CpG island methylation and its reversal by PTS, MSRE-qPCR was performed. The CpG islands of Nrf2 were identified using EMBOSS CpG Plot, revealing two prone CpG islands (394 bp and 1055 bp) with a CG ratio of 60%. Subsequently, primers for Nrf2 targets were designed for CpG island 1, ensuring at least one restriction site within the primer region and no flanking region of the CpG island at the 5′ end.

From the MSRE-qPCR analysis, an increase in methylation percentage of 14.76% in the Nrf2 CpG island was observed in HGM-induced endothelial cells compared to untreated control cells, which exhibited 8.45% basal methylation (Figure 8). Treatment with PTS (10 µM) resulted in a reduction in the methylation percentage to 9.64% in HGM-induced endothelial cells, indicating that PTS acts as a compound that accelerates epigenetic reactivation.

Furthermore, a strong negative correlation was observed between the PTS-pretreated HGM group and Nrf2 expression in endothelial cells pretreated with PTS (r = −0.999; *p* = 0.028). This suggests that the demethylation of the CpG island of the Nrf2 promoter by PTS contributes to the epigenetic reactivation of Nrf2 and enhances wound healing.

## 4. Discussion

The impaired wound healing observed in prolonged hyperglycemia, attributed to ROS accumulation, underscores the significance of Nrf2 in preserving endothelial cell function [4]. Under normal conditions, Nrf2 resides in the cytoplasm bound to its negative regulator, Keap1. Upon cellular stress, Nrf2 dissociates from Keap1, translocates to the nucleus, and activates antioxidant and detoxifying genes [36]. However, it has been revealed that prolonged hyperglycemia alters Nrf2 expression, delaying wound healing [37]. The activation of Nrf2 by natural compounds has gained interest in recent years. Studies have revealed the efficacy of numerous natural compounds to reduce cellular stress and activate Nrf2 including quercetin, morin, naringenin, vitexin, mangiferin, and epicatechin [38,39].

Pterostilbene (PTS), a compound belonging to the stilbene family, possesses antioxidant activity due to the presence of two methoxy groups in its side chain. Unlike its structural analogue resveratrol, PTS is noted for its high absorption and bioavailability [21]. Additionally, PTS has been extensively studied for its high membrane permeability, attributed to its low polar surface area and lipophilicity [40].

Previous studies have accumulated evidence regarding the ability of PTS to activate Nrf2, particularly in nuclear translocation and activation in macrophages through downstream mediator HO-1 [29]. As anticipated by Bhakkiyalakshmi et al., the mechanism of Nrf2 activation by PTS involves preventing its binding to Keap1, a negative regulator, thereby bypassing the ubiquitination process [41]. Furthermore, PTS has been shown to possess antiglycemic activity [42] and antioxidant potential by reducing oxidative stress in the kidney and liver cells of streptozotocin-induced diabetic rats [43].

The mode of Nrf2 activation by PTS has been elucidated in several studies. The activation of Nrf2 occurs via phosphorylation by various kinases such as AKT (protein kinase B), JNK (c-Jun *N*-terminal kinase), PKC (protein kinase C), PI3K (phosphatidylinositol 3-kinase), and ERK (extracellular-signal-regulated kinase), leading to dissociation from Keap1 and subsequent translocation [44]. The modification of cysteine residues in Nrf2 induces conformational changes in Keap1, facilitating Nrf2 dissociation and translocation [45]. Cysteine residues within Nrf2 play a crucial role in maintaining redox homeostasis by suppressing ubiquitination–proteasomal degradation by Keap1 [46]. Another mode of Nrf2 activation involves electrophile-mediated inhibition of Nrf2 ubiquitination–proteasomal degradation [47]. Additionally, studies have shown that PTS binds to the R triad active site of Keap1, facilitating dissociation from Keap1 and subsequent translocation [48]. While the potential of PTS in activating Nrf2 has been established, its mechanism of action in the epigenetic regulation of Nrf2 remains to be elucidated.

In our investigation, we aimed to explore the epigenetic regulation of Nrf2 by PTS in endothelial cells under diabetic conditions, where the dysregulation of Nrf2 signaling is associated with delayed wound healing. To mimic the diabetic environment, we utilized high-glucose media (HGM) at a concentration of 33.3 mM, along with a cytokine cocktail, which has been shown to closely mimic human diabetic conditions [24,49,50].

Initially, we assessed the cytotoxicity of PTS using the Alamar blue assay on endothelial cells exposed to various concentrations of PTS at two different time points (24 and 48 h). We observed no toxic effects at lower concentrations of PTS (0–10 µM) at both time points. However, higher concentrations (25–100 µM) resulted in toxicity, with more than a 50% reduction in cell viability observed at 100 µM after both 24 and 48 h of exposure. This time-dependent cytotoxicity of PTS highlights the importance of concentration and exposure duration in its effects on endothelial cells.

To further evaluate the cytotoxicity of PTS under hyperglycemic conditions, we examined concentrations ranging from 2 to 25 µM. Treatment with PTS at 25 µM resulted in a more than 50% reduction in cell number. Conversely, the cell viability of PTS-treated cells under hyperglycemic conditions at concentrations of 2–10 µM showed an increase compared to the HGM-exposed group. Based on these findings, we fixed the concentration of PTS to 10 µM for further experiments. Our experimental setup, including the concentration of PTS and the duration of exposure to an HGM, aligns with previous studies conducted in macrophage cells by Goutham et al. [29] and with our own previous reports [24]. These studies used similar PTS concentrations and exposure times to mimic diabetic conditions, providing a consistent framework for our investigation.

Nrf2 plays a crucial role as a regulator of redox homeostasis, activating antioxidant enzymes that counteract cellular stress. Studies have associated dysregulated cellular stress to impaired wound healing in diabetes [51]. Notably, Aleksunes et al. found that Nrf2 knockout mice exhibited lower serum insulin levels and exacerbated diabetes due to prolonged hyperglycemia [52]. In our investigation, we observed a reduction in Nrf2 expression in HGM-induced endothelial cells, which was countered by PTS treatment. Furthermore, PTS treatment under hyperglycemic conditions led to an increase in Nrf2 protein levels, indicating PTS-mediated Nrf2 activation and its protective role against prolonged hyperglycemia. This upregulation of Nrf2 expression in PTS-exposed HGM-induced endothelial cells underscores the protective effect of PTS in a diabetic environment.

The activation of Nrf2 triggers the expression of downstream target genes such as HO-1, SOD, CAT, and NQO1, crucial for combating oxidative stress. These genes, with aberrant expression under HGM conditions, were augmented with PTS pretreatment. Specifically, SOD2, a key enzyme involved in reducing mitochondrial ROS accumulation, showed increased expression in PTS-pretreated endothelial cells under HGM conditions. This upregulation of SOD2 expression could contribute to mitigating ROS production in HGM-induced endothelial cells, potentially aiding in the recovery process. Bellot et al. investigated the activation of SOD2 triggered by negative pressure wound therapy (NPWT), which enhanced wound healing by inhibiting ROS production in participants with diabetes [53].

In our current study, we observed an increase in the expression of the SOD2 gene in PTS-pretreated endothelial cells under HGM conditions. This upregulation of SOD2 expression suggests a potential role for PTS in mitigating ROS production in HGM-induced endothelial cells, which could aid in the recuperation process. Previous research has indicated that dysregulation in Nrf2 signaling may arise from genetic alterations or, in certain instances, single nucleotide polymorphisms (SNPs). For instance, we have highlighted the association of the SNP rs182428269 with the pathogenesis of DFU [54].

However, beyond SNPs, hereditable changes without nucleotide alterations, known as epigenetic modifications, also play a role in Nrf2 signaling dysregulation and disease progression. Epigenetic mechanisms such as HDAC modification and DNA methylation contribute to gene silencing, thus altering disease pathogenesis. Abnormal methylation patterns in gene promoter regions have also been implicated in disease progression and impaired wound healing [55]. Small molecules like curcumin, resveratrol, genistein, and quercetin have been reported to alter the expression of epigenetic markers [56].

In the present study, we aimed to investigate the potential of PTS in altering these epigenetic hallmarks. Our gene expression analysis revealed a decrease in the expression of HDAC1/3 and 4 in cells treated with PTS in a diabetic microenvironment. Conversely, the lowered expression of HDAC2 in HGM-induced endothelial cells was upregulated when the cells were pretreated with PTS. Earlier studies have shown that the expression of class 1 HDAC 1/2 and 3 is downregulated in Nrf2^-/-^ mice models, indicating an association between Nrf2 and HDAC expression [57].

Supporting this, another study recorded a decrease in HDAC2 expression, impairing Nrf2 signaling and leading to oxidative stress [58]. Additionally, the inhibition of HDAC by trichostatin A activates Nrf2, protecting against ischemic stroke [59]. The dysregulation of HDAC4 expression has been shown to affect Nrf2 expression, as evidenced by upregulated HDAC4 expression activating NF-kB signaling, which mediates Nrf2 suppression [60]. To support our findings on HDAC3 inhibition, Huang et al. highlighted that inhibiting HDAC3 expression protects against T2DM-induced endothelial dysfunction through Nrf2 activation [61].

The epigenetic-mediated silencing of the promoter region of target genes by DNA methyltransferase enzymes contributes to transcriptional repression [62]. Studies on the epigenetic activation of Nrf2 have shown that DNA demethylation in the CpG island of the target gene promoter plays a pivotal role in cellular homeostasis. In our study, we found that the expression of DNMTs (DNMT1, 3A, and 3B) was profoundly higher in endothelial cells exposed to HGM conditions, thus inhibiting Nrf2 expression. Similarly, aberrant expression of DNMTs reduces Nrf2 expression, corresponding to disease progression [63]. Another study by Zhang et al. stated that altered methylation patterns in the Nrf2 promoter region by DNMT enzyme activity contribute to oxidative stress-induced pathogenesis [64].

Epigenetic alteration in the Nrf2 promoter region can be reversed or halted by the action of small molecules [65]. Furthermore, it has been revealed that PTS has the ability to reduce the methylation of the Nrf2 CpG promoter in endothelial cells pretreated with PTS under HGM conditions compared to cells exposed only to an HGM. From the MSRE-PCR results, it was evident that PTS reversed the methylation from 14% to 9% in the PTS-treated group. Our results highlight the effective role of PTS in hypomethylating the gene promoter, thereby activating Nrf2 and maintaining redox homeostasis.

To the best of our knowledge, we are the first to report on the epigenetic reversal potential of PTS by demethylating the CpG island of the Nrf2 promoter region. Supporting this, a study conducted by Khor et al. demonstrated that the demethylation of the Nrf2 promoter region may contribute to the prevention of antioxidant stress in human prostate cancer [63]. Another study by Lou et al. highlighted that the demethylation of the Nrf2 promoter region protects against nano-SiO_2_-induced carcinogenesis in human bronchial epithelial cells [66]. Collectively, we emphasize the potential role of PTS in the epigenetic reactivation of Nrf2 by inhibiting the action of epigenetic hallmarks and helping maintain cellular homeostasis.

## 5. Conclusions

Our findings from this study suggest that Nrf2 is a potential target for combating impaired wound healing through epigenetic regulation. The natural stilbene PTS facilitates the epigenetic reactivation of Nrf2, indicating its potential as a potent epigenetic modulator for enhancing wound healing in diabetic conditions. Specifically, PTS decreases the expression of HDACs and DNMTs, facilitating Nrf2 promoter demethylation and contributing to Nrf2 reactivation. This reactivation of Nrf2 triggers the transcriptional activation of antioxidant response element genes, including SOD2, which combats ROS-mediated stress and promotes wound healing in a diabetic environment. Additionally, the ability of PTS to modulate these epigenetic mechanisms highlights its significant therapeutic potential. In conclusion, our study offers a potential therapeutic approach for improving wound healing, suggesting that PTS could be developed as a drug targeting epigenetic regulation to enhance healing outcomes in diabetic patients.

## Figures and Tables

**Figure 1 nutrients-16-02045-f001:**
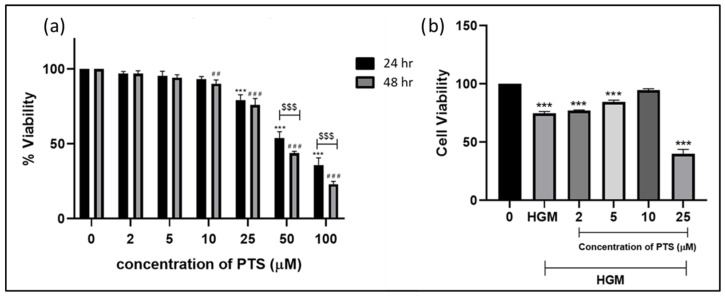
Dose-dependent toxicity of pterostilbene (PTS) on endothelial cells at different time points as assessed by Alamar blue assay (**a**) and in a hyperglycemic microenvironment for 48 h (**b**). Data are represented as mean ± SEM. * Significance compared to 24 h control group; ^#^ significance compared to 48 h control group; ^$^ significance compared within the time points among groups. ^##^
*p* < 0.01, and ***^###$$$^
*p* < 0.001.

**Figure 2 nutrients-16-02045-f002:**
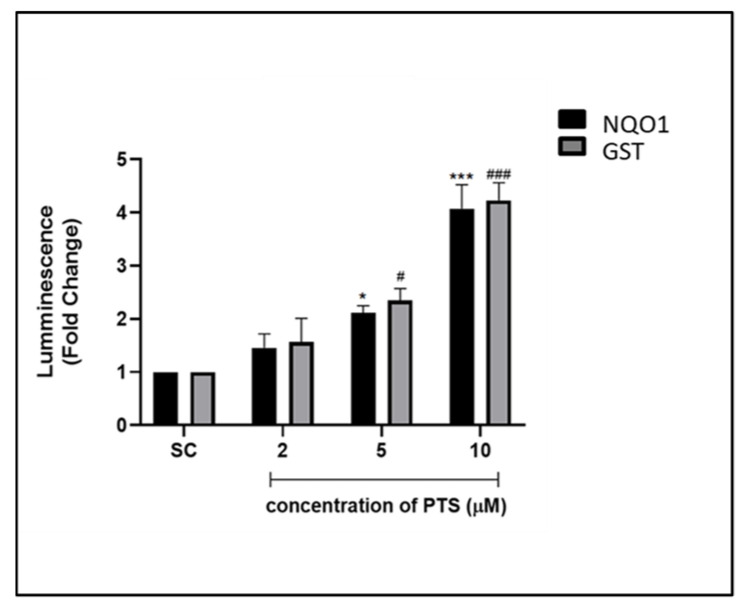
Nrf2 activation potential of various concentrations of PTS on endothelial cells as determined by reporter assay using NQO1 and GST1-ARE-Luc gene constructs. Data are represented as mean ± SEM. * Significance compared with control; ^#^ significance compared with HGM; SC—scrambled control. *^#^
*p* < 0.05, and ***^###^
*p* < 0.001.

**Figure 3 nutrients-16-02045-f003:**
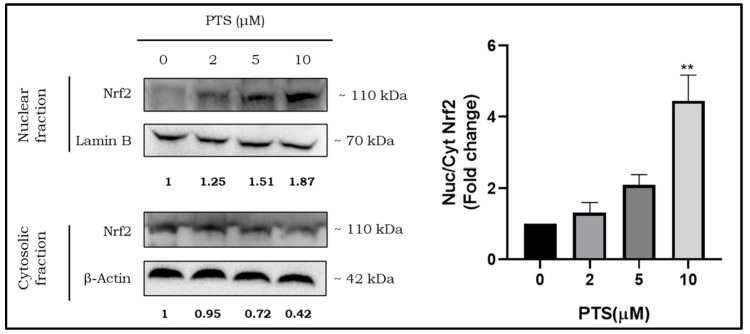
Effect of PTS on Nrf2 translocation in endothelial cells. The nuclear/cytosolic ratio fold change of Nrf2 translocation is represented as a bar diagram. Data are represented as mean ± SEM. PTS—pterostilbene; kDa—kilodalton; Nuc—nuclear; Cyt—cytosolic. ** *p* < 0.01.

**Figure 4 nutrients-16-02045-f004:**
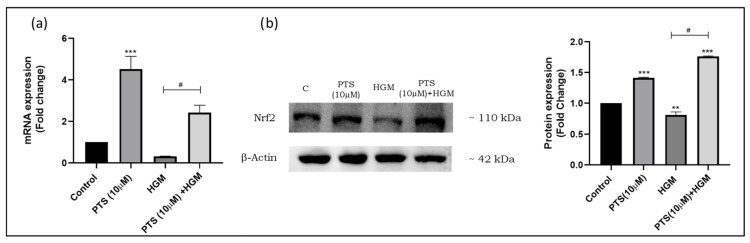
Effect of PTS on mRNA expression (**a**) and protein expression of Nrf2 (**b**) in HGM-induced endothelial cells as assessed by qRT-PCR and Western blot analysis. Data are represented as mean ± SEM. * Significance compared with control; ^#^ significance compared with HGM. *^#^
*p* < 0.05, ** *p* < 0.01, and *** *p* < 0.001.

**Figure 5 nutrients-16-02045-f005:**
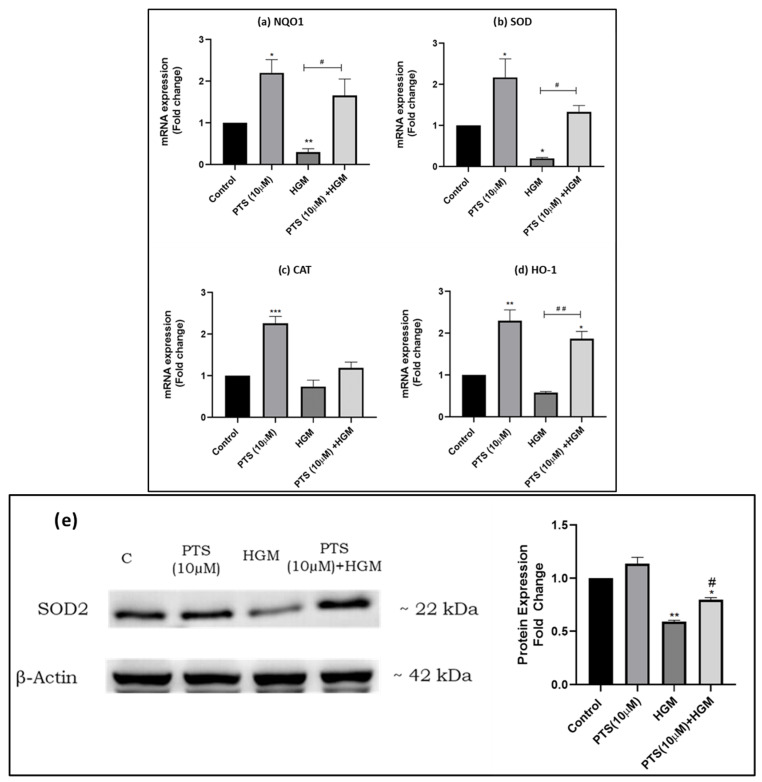
(**a**–**d**) Effect on PTS on mRNA expression of Nrf2 downstream targets NQO1 (**a**), SOD (**b**), CAT (**c**), and HO-1 (**d**) in HGM-induced endothelial cells. (**e**) Effect of PTS on protein expression of SOD2 in HGM-exposed endothelial cells as assessed by Western blot analysis. Data are represented as mean ± SEM. * Significance compared with control; ^#^ significance compared with HGM. *^#^
*p* < 0.05, **^##^
*p* < 0.01, and *** *p* < 0.001.

**Figure 6 nutrients-16-02045-f006:**
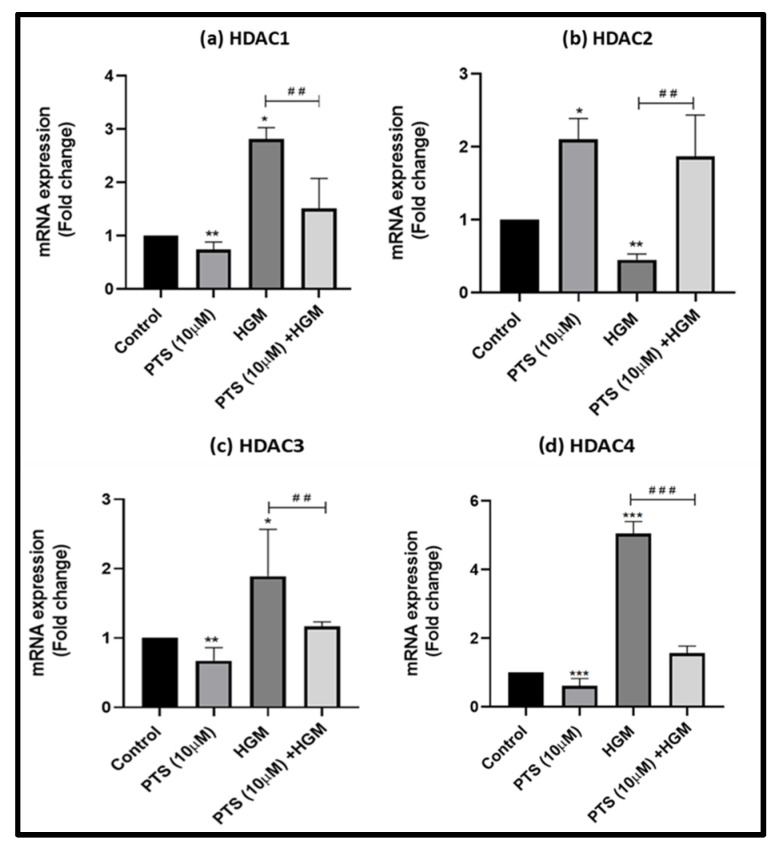
Effect of PTS on the mRNA expression of epigenetic erasers HDAC1 (**a**), HDAC2 (**b**), HDAC3 (**c**), and HDAC4 (**d**) in HGM-induced endothelial cells. Data are represented as mean ± SEM. * Significance compared with control; ^#^ significance compared with HGM. * *p* < 0.05, **^##^
*p* < 0.01, and ***^###^
*p* < 0.001.

**Figure 7 nutrients-16-02045-f007:**
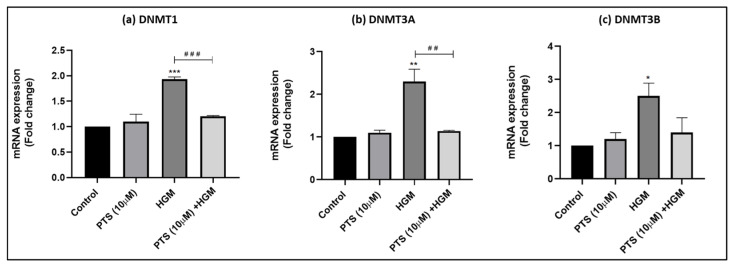
Effect of PTS on the mRNA expression of epigenetic writers DNMT 1 (**a**), DNMT3A (**b**), and DNMT3B (**c**) in endothelial cells exposed to HGM conditions. Data are represented as mean ± SEM. * Significance compared with control; ^#^ significance compared with HGM. * *p* < 0.05, **^##^
*p* < 0.01, and ***^###^
*p* < 0.001.

**Figure 8 nutrients-16-02045-f008:**
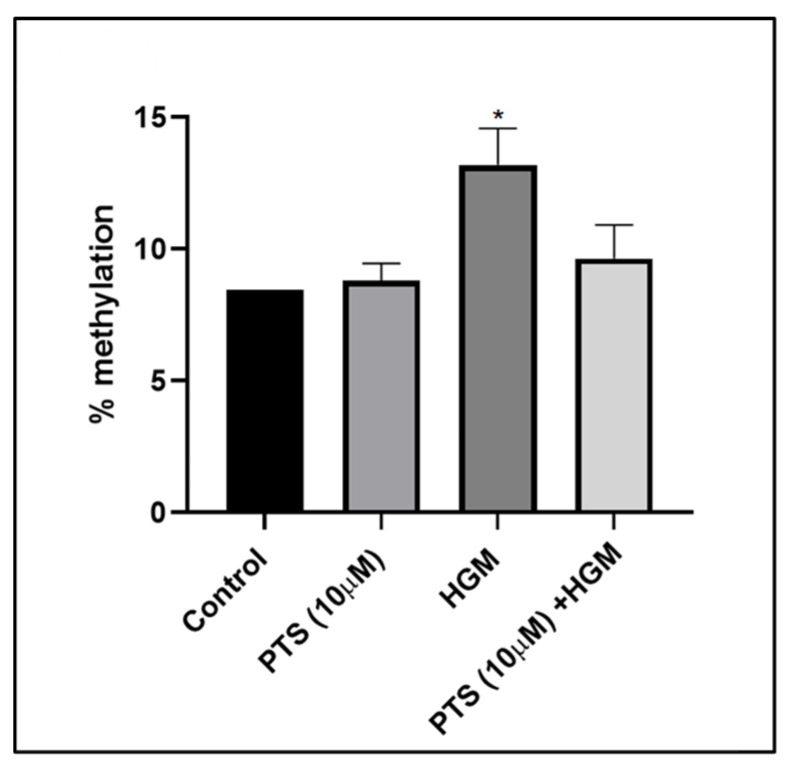
Effect of PTS on epigenetic reversal in DNA demethylation in CpG island of Nrf2 promotor region represented as percentage as assessed by MSRE-qPCR among PTS-pretreated endothelial cells exposed to HGM conditions. * Significance compared with control; ^#^ significance compared with HGM. * *p* < 0.05.

**Table 1 nutrients-16-02045-t001:** List of primers and their sequences used in this study.

S. No.	Gene	Forward Primer	Reverse Primer	Tm (°C)
1	Nrf2	TGTAGATGACAATGAGGTTTC	ACTGAGCCTGATTAGTAGCAA	56 °C
2	NQO1	AGGATGGAAGAAACGCCTGG	TCAGTTGGGATGGACTTGCC	60 °C
3	SOD	GGCATCATCAATTTCGAG	CCGTAGTAGTTAAAGCTC	59 °C
4	CAT	ATCCGTGTAACCCGCTCATC	ACCTTCATTTTCCCCTGGGG	61 °C
5	HO-1	GGGAATTCTCTTGGCTGGCT	AACTGAGGATGCTGAAGGGC	59 °C
6	HDAC1	GGCTGGCAAAGGCAAGTAT	CGCACTAGGCTGGAACATCT	58 °C
7	HDAC2	ATTGGGGAACAGGTGGTG	GGGGCGAGGGATAAAAGA	56 °C
8	HDAC3	GTATGAAGTCGGGGCAGAGA	CGTGGGTTGGTAGAAGTCC	55.5 °C
9	HDAC4	GCACAGTCCTTGGTTGGT	AGAAACTGCTGATGCTGCT	56 °C
10	DNMT1	TCAAGACTGATGGGAAGAAGAGTT	CGTGACCCTTGCTAGATACAGC	56 °C
11	DNMT3A	GATGACGAGCCAGAGTACGA	CTTCTCAACACACACCACTGA	56 °C
12	DNMT3B	CGACCTCACAGACGACACAG	TCCAAACTCCTTCCCATCCT	56.2 °C
13	GAPDH	AAGAAGGTGGTGAAGCAGGC	GTCAAAGGTGGAGGAGTGGG	60 °C

## Data Availability

The raw data supporting the conclusions of this article will be made available by the authors on request.

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
