# Peer review of "Pterostilbene Reverses Epigenetic Silencing of Nrf2 and Enhances Antioxidant Response in Endothelial Cells in Hyperglycemic Microenvironment"

_nutrients, 2024, doi:10.3390/nu16132045_

Round 1
Reviewer 1 Report
Comments and Suggestions for Authors
The text below contains comments on manuscript entitled “Pterostilbene Reverses Epigenetic Silencing of Nrf2 and Enhances Antioxidant Response in Endothelial Cells under Hyperglycemic Microenvironment”
The manuscript is focused to investigate the effect of pterostilbene (PTS) on the epigenetic reactivation of Nrf2 in endothelial cells under a hyperglycaemic microenvironment. The manuscript is well written on sufficiently good English. The experimental design is logically structured and the obtained results fully explain the aim of the study.
I have some minor suggestions for corrections:
Page 3, Line 93: Please add the catalogue number and purity of pterostilbene.
To my opinion, the authors should focus more on the conclusions and strengthen this part.
Comments on the Quality of English LanguageMinor editing of English language required
Author Response
The manuscript is focused to investigate the effect of pterostilbene (PTS) on the epigenetic reactivation of Nrf2 in endothelial cells under a hyperglycaemic microenvironment. The manuscript is well written on sufficiently good English. The experimental design is logically structured and the obtained results fully explain the aim of the study.
Comment: Page 3, Line 93: Please add the catalogue number and purity of pterostilbene
Response: As per the reviewer’s suggestion, we have included the catalogue number and purity of the pterostilbene and highlighted the same in the revised manuscript Page 3, Line 99.
Comment: To my opinion, the authors should focus more on the conclusions and strengthen this part.
Response: As per the reviewer's comment, we have restructured the conclusion and highlighted the same in the revised manuscript Page 13, Line 495-506.
Reviewer 2 Report
Comments and Suggestions for Authors
The manuscript is well written, and the aims of the review are clear.
The manuscript has very long sections. Therefore, I recommend the Authors for easier reading to shorten the sections by dividing it into paragraphs.
In addition, where possible, the inclusion of figures or diagrams could make the review more explanatory
Comments on the Quality of English LanguageEnglish Language is good
Author Response
Comment: The manuscript has very long sections. Therefore, I recommend the Authors for easier reading to shorten the sections by dividing it into paragraphs.
Response: As per the reviewer’s suggestion, we have divided the long sections into small paragraphs in the revised manuscript.
Comment - In addition, where possible, the inclusion of figures or diagrams could make the review more explanatory
Response: As per the reviewer’s comments, we have split the figure 2 (Fig 2a and Fig 2b) and figure 4 (fig 4(a-d) and fig 4e) and cited at the relevant text, and highlighted the same in the revised manuscript.
Reviewer 3 Report
Comments and Suggestions for Authors
The manuscript "Pterostilbene Reverses Epigenetic Silencing of Nrf2 and Enhances Antioxidant Response in Endothelial Cells under Hyperglycemic Microenvironment " fits the journal's scope. The authors present their results on the regulation of nuclear factor erythroid 2-related factor 2 (Nrf2) by pterostilbene.
The research design is detailed and clearly presented and the novelty is highlighted.
The experiments are described in sufficient detail in the Materials and Methods section; however, the relevant references are missing. Please add these references to this section.For Section 2.9, please check if the software tools used require citation.
The results are clearly presented and well-organized, with relevant figures and graphs included. Although some information is repeated from the Materials and Methods section, this repetition ensures clarity in the research design.
The results are further discussed in relation to other researchers' findings in the Discussion section, providing arguments for each step of the research. Despite being somewhat extensive, the discussion contributes to the clarity and facilitates the reading of the manuscript.
The conclusion is supported by the results obtained from the experiments; however, this section could be expanded.
Other remarks:
The importance of pterostilbene should be more thoroughly presented in the Introduction. Additionally, a phrase regarding the sources of this compound should be included.
Lines 85, 234, 225 - please correct the errors
Figure 7 b is not necessary and should be removed.
Author Response
Comment: The experiments are described in sufficient detail in the Materials and Methods section; however, the relevant references are missing. Please add these references to this section. For Section 2.9, please check if the software tools used require citation.
Response: As per the reviewer’s suggestion, we have updated the materials and methods section with relevant references and highlighted in the revised manuscript.
Comment - The results are clearly presented and well-organized, with relevant figures and graphs included. Although some information is repeated from the Materials and Methods section, this repetition ensures clarity in the research design.
Response: We authors of this article, gratefully acknowledge the reviewer's time to conduct details review and comments.
Comment: The results are further discussed in relation to other researchers' findings in the Discussion section, providing arguments for each step of the research. Despite being somewhat extensive, the discussion contributes to the clarity and facilitates the reading of the manuscript.
Response: Thank you for your positive feedback on the Discussion section. We aimed to provide thorough comparisons with existing research to ensure a robust analysis and to enhance the reader's understanding of our findings.
Comment: The conclusion is supported by the results obtained from the experiments; however, this section could be expanded.
Response: Thank you for your positive feedback on the conclusion section. We have expanded the conclusion section to provide a more comprehensive summary of our findings, discuss the implications of our results in greater detail, and outline potential directions for future research.
Other remarks:
Comment: The importance of pterostilbene should be more thoroughly presented in the Introduction. Additionally, a phrase regarding the sources of this compound should be included.
Response: As per the reviewer’s suggestion, we have added the importance and sources of PTS in the introduction section and highlighted the same in the revised manuscript.
Comment: Lines 85, 234, 225 - please correct the errors
Response: We regret the errors encountered. As per the reviewer’s comments, we have addressed the error and corrected in the revised manuscript and highlighted the same.
Comment: Figure 7 b is not necessary and should be removed.
Response: As per the reviewer’s comments, we have removed Figure 7b in the revised manuscript.